# Make Continual Learning Stronger via C-Flat

**Ang Bian**[*1], **Wei Li**[*1,2], **Hangjie Yuan** [3,4], **Chengrong Yu**[1], **Mang Wang**[5]
**Zixiang Zhao**[6], **Aojun Lu**[1], **Pengliang Ji**[7], **Tao Feng**[†2]

[1]Sichuan University     [2]Tsinghua University     [3]DAMO Academy, Alibaba Group
[4]Zhejiang University     [5]ByteDance     [6]Xi'an Jiaotong University     [7]Carnegie Mellon University
hj.yuan@zju.edu.cn, {ymjiii98, fengtao.hi}@gmail.com

## Abstract

How to balance the learning 'sensitivity-stability' upon new task training and memory preserving is critical in CL to resolve catastrophic forgetting. Improving model generalization ability within each learning phase is one solution to help CL learning overcome the gap in the joint knowledge space. Zeroth-order loss landscape sharpness-aware minimization is a strong training regime improving model generalization in transfer learning compared with optimizer like SGD. It has also been introduced into CL to improve memory representation or learning efficiency. However, zeroth-order sharpness alone could favors sharper over flatter minima in certain scenarios, leading to a rather sensitive minima rather than a global optima. To further enhance learning stability, we propose a **C**ontinual **Flat**ness (**C-Flat**) method featuring a flatter loss landscape tailored for CL. C-Flat could be easily called with only one line of code and is plug-and-play to any CL methods. A general framework of C-Flat applied to all CL categories and a thorough comparison with loss minima optimizer and flat minima based CL approaches is presented in this paper, showing that our method can boost CL performance in almost all cases. Code is available at `https://github.com/WanNaa/C-Flat`.

**C-Flat**: just **a line of code** suffices for its utilization.

```
1  from ... import C_Flat
2
3  # initialize optimizer for CL.
4  C_Flat_optimizer = C_Flat(params, base_optimizer, model, args)
```

## 1   Introduction

**Why study Continual Learning (CL)?** CL is generally acknowledged as a necessary attribute for Artificial General Intelligence (AGI) [22, 55, 40, 67]. In the open world, CL holds the potential for substantial benefits across many applications: *e.g.* vision model needs to learn a growing image set [17, 61, 62], or, embodied model needs to incrementally add skills to their repertoire [12].

**Challenges.** A good CL model is expected to keep the memory of all seen tasks upon learning new knowledge [22]. However, due to the limited access to previous data, the learning phase is naturally sensitive to the current task, hence resulting in a major challenge in CL called catastrophic forgetting [9], which refers to the drastic performance drop on past knowledge after learning new knowledge. This learning sensitivity-stability dilemma is critical in CL, requiring model with strong generalization ability [16] to overcome the knowledge gaps between sequentially arriving tasks.

---

[*]Equal Contribution
[†]Corresponding Authors

38th Conference on Neural Information Processing Systems (NeurIPS 2024).

**Current solutions.** A series of works [43, 44, 33, 25] are proposed to improve learning stability by extending data space with dedicated selected and stored exemplars from old tasks, or frozen some network blocks or layers that are strongly related to previous knowledge [68, 24, 69, 57, 24].

Another group of works seeks to preserve model generalization with regularisation onto the training procedure itself [32, 18, 31]. Diverse weight [45, 28, 2] or gradient alignment [22, 9, 35, 26] strategies are designed to encourage the training to efficiently extracting features for the current data space without forgetting.

**Loss landscape sharpness** optimization [23, 19, 65, 70] as an efficient training regime for model generalization starts to gain attentions [27, 63]. Ordinary loss minima based optimizer like SGD can easily lead to suboptimal results [4, 37, 13]. To prevent this, zeroth-order sharpness-aware minimization seeking neighborhood-flat minima [20] has been proven a strong optimizer to improve model generalization ability, especially in transferring learning tasks. It is also introduced into some CL works [49, 30] with dedicated designs to improve old knowledge representation or few-shot learning efficiency. However, given the limited application scenarios[10, 49], the zeroth-order sharpness used in the current work is proved to favor sharper minima than a flat solution [70]. It means zeroth-order only can still lead to a fast gradient descent to the suboptimal in new data space than a more generalizable result for the joint old and new knowledge space.

**Our solution.** Inspired by these works, a beyond zeroth-order sharpness continual optimization method is proposed as demonstrated in 1, where loss landscape flatness is emphasized to strengthen model generalization ability. Thus, the model can always converge to a flat minima in each phase, and then smoothly migrate to the global optimal of the joint knowledge space of the current and next tasks, and hence resolve the catastrophic forgetting in CL. We dub this method **C**ontinual **Flat**ness (C-Flat or $C\flat$) Moreover, C-Flat is a general method that can be easily plug-and-play into any CL approach with only one line of code, to improve CL.

**Contribution.** A simple and flexible CL-friendly optimization method C-Flat is proposed, which Makes Continual Learning Stronger.

A framework of C-Flat covering diverse CL method categories is demonstrated. Experiment results prove that Flatter is Better in nearly all cases.

To the best of our knowledge, this work is the first to conduct a thorough comparison of CL approaches with loss landscape aware optimization, and thus can serve as a baseline in CL.

## 2   Related work

**Continual learning** methods roughly are categorized into three groups: *Memory-based* methods write experience in memory to alleviate forgetting. Some work [43, 44, 25, 51] design different sampling strategies to establish limited budgets in a memory buffer for rehearsal. However, these methods require access to raw past data, which is discouraged in practice due to privacy concerns. Instead, recently a series of works [10, 34, 46, 33, 50] elaborately construct special subspace of old tasks as the memory. *Regularization-based* methods aim to realize consolidation of the previous knowledge by introducing additional regularization terms in the loss function. Some works [32, 29, 6] enforce the important weights in the parameter space [45, 28, 2], feature representations [5, 21], or the logits outputs [32, 42] of the current model function to be close to that of the old one. *Expansion-based* methods dedicate different incremental model structures towards each task to minimize forgetting [68, 39]. Some work [48, 24, 60] exploit modular network architectures (dynamically extending extra components [57, 69], or freeze partial parameters [36, 1]) to overcome forgetting. Trivially, methods in this category implicitly shift the burden of storing numerous raw data into the retention of model [68].

**Gradient-based solutions** are a main group in CL, including shaping loss landscape, tempering the tug-of-war of gradient, and other learning dynamics [22, 9, 41]. One promising solution is to modify the gradients of different tasks and hence overcome forgetting [7, 38], e.g., aligning the gradients of current and old one [15, 18], or, learning more efficient in the case of conflicting objectives [47, 56, 14]. Other solutions [10, 49] focus on characterizing the generalization from the loss landscape perspectives to improve CL performance and yet are rarely explored.

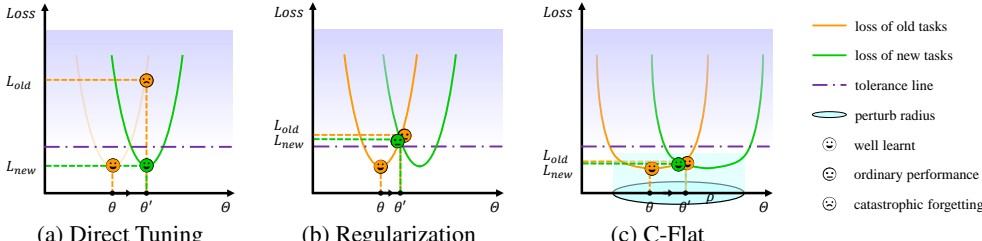

Figure 1: Illustration of C-Flat overcoming catastrophe forgetting by fine-tuning the old model parameter to flat minima of new task. a) loss minima for current task only can cause catastrophe forgetting on previous ones. b) balanced optima aligned by regularization leads to unsatisfying results for both old and new tasks. c) C-Flat seeks global optima for all tasks with flattened loss landscape.

**Sharpness minimization in CL** Many recent works [23, 19, 4] are proposed to optimize neural networks in standard training scenarios towards flat minima. Wide local minima were considered an important regularization in CL to enforce the similarity of important parameters learned from past tasks [6]. Sharpness-aware seeking for loss landscape flat minima starts to gain more attention in CL, especially SAM based zeroth order sharpness is well discussed. An investigation [41] proves SAM can help with addressing forgetting in CL, and [8] proposed a combined SAM for few-shot CL. SAM is also used for boosting the performance of specific methods like DFGP [58] and FS-DGPM [10] designed for GPM. SAM-CL [52] series with loss term gradient alignment for memory-based CL. These efforts kicked off the study of flat minima in CL, however, zeroth-order sharpness may not be enough for flatter optimal [70]. Thus, flatness with a global optima and universal CL framework is further studied.

## 3  Method

Our solution addresses the learning sensitivity-stability dilemma in CL by improving model generalization for joint learning knowledge obtained from different catalogues domains or tasks. Moreover, a general but stronger optimization method enhanced by the latest gradient landscape flatness is proposed as a 'plug-and-play' tool for any CL approach.

**Loss landscape flatness.** Let $B(\theta, \rho) = \{\theta' : \|\theta' - \theta\| < \rho\}$ denotes the neighborhood of $\theta$ with radius $\rho > 0$ in the Euclidean space $\Theta \subset \mathbb{R}^d$, the zeroth-order sharpness at point $\theta$ is commonly defined by the maximal training loss difference within its neighborhood $B(\theta, \rho)$:

$$R_\rho^0(\theta) = \max\{\ell_S(\theta') - \ell_S(\theta) : \theta' \in B(\theta, \rho)\}. \tag{1}$$

where $\ell_S(\theta)$ denotes the loss of an arbitrary model with parameter $\theta$ on any dataset $S$ with an oracle loss function $\ell(\cdot)$. The zeroth-order sharpness $R_\rho^0(\theta)$ regularization can be directly applied to restrain the maximal neighborhood training loss:

$$\ell_S^{R_\rho^0}(\theta) = \ell_S(\theta) + R_\rho^0(\theta) = \max\{\ell_S(\theta') : \theta' \in B(\theta, \rho)\}, \tag{2}$$

However, for some fixed $\rho$, local minima with a lower loss does not always have a lower major hessian eigenvalue [70], which equals to the neighborhood curvature. It means that zeroth-order sharpness optimizer may goes to a sharper suboptimal than to the direction of a flatter global optimal with better generalization ability.

Recently, first-order gradient landscape flatness is proposed as a measurement of the maximal neighborhood gradient norm, which reflects landscape curvature, to better describe the smoothness of the loss landscape:

$$R_\rho^1(\theta) = \rho \cdot \max\{\|\nabla \ell_S(\theta')\|_2 : \theta' \in B(\theta, \rho)\}. \tag{3}$$

Unlike zeroth-order sharpness that force the training converging to a local minimal, first-order flatness alone constraining on the neighborhood smoothness can not lead to an optimal with minimal loss. To maximize the generalization ability of loss landscape sharpness for continual learning task, we

propose a zeroth-first-order sharpness aware optimizer C-Flat for CL. Considering the data space, model or blocks to be trained are altered regarding the training phase and CL method, (as detailed in the next subsection), we define the the C-Flat loss as follows:

$$\ell_{S^T}^C(f^T(\theta^T)) = \ell_{S^T}(f^T(\theta^T)) + R_{\rho,S^T}^0(f^T(\theta^T)) + \lambda \cdot R_{\rho,S^T}^1(f^T(\theta^T))$$

$$= \ell_{S^T}^{R_\rho^0}(f^T(\theta^T)) + \lambda \cdot R_{\rho,S^T}^1(f^T(\theta^T)), \tag{4}$$

with the minimization objective:

$$\min_{\theta^T}\{\max\{\ell_{S^T}(f^T(\theta_0^T)) + \lambda\rho \cdot \|\nabla\ell_{S^T}(f^T(\theta_1^T))\|_2\} : \theta_0^T, \theta_1^T \in B(\theta^T, \rho)\} \tag{5}$$

where $\ell_S^{R_\rho^0}(\theta)$ is constructed to replace the original CL loss, while $R_\rho^1(\theta)$ further regularizes the smoothness of the neighborhood, and hyperparameter $\lambda$ is to balance the influence of $R_\rho^1$ as an additional regularization to loss function $\ell$. Hence, the local minima within a flat and smooth neighborhood is calculated for a generalized model possessing both old and new knowledge.

**Optimization.** In our work, the two regularization terms in the proposed C-Flat are resolved correspondingly in each iteration. Assuming the loss function $\ell(\cdot)$ is differentiable and bounded, the gradient of $\ell_S^{R_\rho^0}$ at point $\theta^T$ can be approximated by

$$\nabla\ell_S^{R_\rho^0}(\theta^T) \approx \nabla\ell_S(\theta_0^T) \text{ with } \theta_0^T = \theta^T + \rho \cdot \frac{\nabla\ell_S(\theta^T)}{\|\nabla\ell_S(\theta^T)\|_2} \tag{6}$$

And the gradient of the first-order flatness regularization $\nabla R_\rho^1(\theta^T)$ can be approximated by

$$\nabla R_\rho^1(\theta^T) \approx \rho \cdot \nabla\|\nabla\ell_S(\theta_1^T)\|_2 \tag{7}$$

$$\text{with} \qquad \theta_1^T = \theta^T + \rho \cdot \frac{\nabla\|\nabla\ell_S(\theta^T)\|_2}{\|\nabla\|\nabla\ell_S(\theta^T)\|_2\|_2}$$

$$\text{where} \qquad \nabla\|\nabla\ell_S(\theta^T)\|_2 = \nabla^2\ell_S(\theta^T) \cdot \frac{\nabla\ell_S(\theta^T)}{\|\nabla\ell_S(\theta^T)\|_2}.$$

The optimization is detailed in Appendix algorithm 1. Note that $\nabla\ell$ is the gradient of $\ell$ with respect to $\theta$ through this paper, and instead of the expensive computation of Hessian matrix $\nabla^2\ell$, Hessian-vector product calculation is used in our algorithm, where the time and especially space complexity are greatly reduced to $o(n)$ using 1 forward and 1 backward propagation. Thus, the overall calculation in one iteration takes 2 forward and 4 backward propagation in total.

**Theoretical analysis.** Given $R_\rho^0(\theta)$ measuring the maximal limit of the training loss difference, the first-order flatness is its upper bound by nature. Denoting $\theta + \epsilon \in B(\theta, \rho)$ the local maximum point, a constant $\epsilon^* \in [0, \epsilon]$ exists according to the mean value theorem that

$$R_\rho^0(\theta) = max\{\ell_S(\theta') - \ell_S(\theta) : \theta' \in B(\theta, \rho)\}$$

$$= \ell_S(\theta + \epsilon) - \ell_S(\theta) = (\nabla\ell_S(\theta + \epsilon^*))^T \cdot \epsilon \leq \|\nabla\ell_S(\theta + \epsilon^*)\|_2 \cdot \|\epsilon\|_2$$

$$\leq max\{\|\nabla\ell_S(\theta')\|_2 : \theta' \in B(\theta, \rho)\} \cdot \rho = R_\rho^1(\theta). \tag{8}$$

Assuming the loss function is twice differentiable, bounded by $M$, obeys the triangle inequality, its gradient has bounded variance $\sigma^2$, and both the loss function and its second-order gradient are $\beta-$Lipschitz smooth, we can prove that, according to [3, 63], C-Flat converges in all tasks with $\eta \leq 1/\beta, \rho \leq 1/4\beta$, and $\eta_i^T = \eta/\sqrt{i}, \rho_i^T = \rho/\sqrt[4]{i}$ for epoch $i$ in any task $T$,

$$\frac{1}{n^T}\sum_{i=1}^{n^T}\mathbb{E}[\|\nabla\ell_{S^T}^C(f^T(\theta^T))\|^2] \leq \frac{2}{n^T}\sum_{i=1}^{n^T}\mathbb{E}[\|\nabla\ell_{S^T}^{R_\rho^0}(f^T(\theta^T))\|^2]$$

$$+\frac{2}{n^T}\sum_{i=1}^{n^T}\mathbb{E}[\|\lambda R_{\rho,S^T}^1(f^T(\theta^T))\|^2] \leq \frac{8M\beta}{\sqrt{n^T}} + \frac{16\sigma^2}{3b\sqrt{n^T}} + \frac{32\lambda^2(2\sqrt{n^T}-1)}{\beta^2 n^T}. \tag{9}$$

where $n^T$ is the total iteration numbers of task $T$, and $b$ is the batch size.

**Upper Bound.** Let $\nabla^2 \ell_S(\theta^*)$ denotes the Hessian matrix at local minimum $\theta^*$, its maximal eigenvalue $\lambda_{max}(\nabla^2 \ell_S(\theta^*))$ is a proper measure of the landscape curvature. The first-order flatness is proven to be related to the maximal eigenvalue of the Hessian matrix as $R_\rho^1(\theta^*) = \rho^2 \cdot \lambda_{max}(\nabla^2 \ell_S(\theta^*))$, thus the C-Flat regularization can also be used as an index of model generalization ability, with the following upper bound:

$$R_\rho^C(\theta^*) = R_\rho^0(\theta^*) + \lambda R_\rho^1(\theta^*) \leq (1 + \lambda)\rho^2 \cdot \lambda_{max}(\nabla^2 \ell_S(\theta^*)). \tag{10}$$

### 3.1 A Unified CL Framework Using C-Flat

This subsection presents an unified CL framework using C-Flat with applications covering Class Incremental Learning (CIL) approaches. To keep focus, the scope of our study is limited in CIL task, which is the most intractable CL scenarios that seek for a lifelong learning model for sequentially arriving class-agnostic data. Most CIL approaches belong to three main families, Memory-based, Regularization-based and Expansion-based methods.

**Memory-based** methods store samples from the previous phases within the memory limit, or produce pseudo-samples by generative approaches to extend the current training data space, thus a memory replay strategy is used to preserve the seen class features with $\hat{S}^T = S^T \cup Sample^{t<T}$. iCaRL is one of the early works. It learns classifiers and a feature representation simultaneously, and preserves the first few most representative exemplars according to Nearest-Mean-of-Exemplars Classification. Thus a loss function $\ell_{\hat{S}^T} = \ell_{\hat{S}^T}^{CE} + \ell_{\hat{S}^T}^{KL}$ combining both cross entropy for the current task and a knowledge distillation loss for the previous classes is introduced to balance the learning sensitivity and model generalization to the previous tasks.

**Solution:** for memory-based method, including, the C-Flat can be easily applied to these scenarios by reconstruct the oracle loss function with its zeroth- and first-order flatness measurement as eq. 11, and trained with algorithm 1 using data set extended with the previous exemplars.

$$\ell_{\hat{S}^T}^C(\theta^T) = \ell_{\hat{S}^T}^{R_\rho^0}(\theta^T) + \lambda \cdot \ell_{\hat{S}^T}^{R_\rho^1}(\theta^T). \tag{11}$$

**Regularization-based** methods seeks for a apply regularization on the model develop to preserver the learnt knowledge. For instance, WA introduces weight aligning on the final inference part to balance the old and new classes. Denoting $\phi$ the feature learning layers of the model, $\psi = [\psi^{old}, \psi^{new}]$ the decision head for all classes consisting of two branches for the old and new seen data classes, the output is corrected to $f(x) = [\psi^{old}(\phi(x)), \gamma \cdot \psi^{new}(\phi(x))]$, where $\gamma$ is the fraction of average norm of $\psi^{old}, \psi^{new}$ of all classes.

Gradient Projection Memory (GPM) is another main regularization based group, introducing explicit align the gradient direction to new knowledge learning. It stores a minimum set of bases of the Core Gradient Space as Gradient Projection Memory, thus gradient steps are only taken in its orthogonal direction to learn the new features without forgetting the core information from the previous phases. FS-DGPM further improves this method by updating model parameter along the aligned orthogonal gradient at the zeroth-order sharpness minima in dynamic GMP space.

**Solution:** for regularization-based approaches, the same plug-and-play strategy can be used to reconstruct the loss function as eq. 11, and optimized by algorithm 1.

**An alternative solution** for the gradient-based methods like GPM and the improved FS-DGPM, is to introduce C-Flat optimization at the gradient alignment stage, so that the orthogonal gradient at a flatter minima is used to ensure that the training can cross over the knowledge gap between different data categories. The implementation of our C-Flat-GPM is detailed in Appendix algorithm 2.

**Expansion-based** methods explicitly construct task-specific parameters to resolve the new class learning and inference problem. For instance, Memory-efficient Expandable Model (Memo) decomposes the embedding module into deep layers and shallow layers that $\phi = \phi_f(\phi_g)$, where $\phi_f, \phi_g$ correspond to the specialized block for different tasks and the generalized block that can be shared during training phases. An additional block $\phi_f^{new}$ is added to the deep layers for specified feature extraction for the new classes, where the model can be reconstructed as $f^T = \psi^T([\phi_f^{T-1}(\phi_g), \phi_f^{new}(\phi_g)])$. Thus the new model training is focusing on the task specified component while the shared shallow layers are frozen with loss function $\ell_{\hat{S}^T}^{Memo} = \ell_{\hat{S}^T}^{CE}(\psi^T([\phi_f^{T-1}(\phi_g), \phi_f^{new}(\phi_g)]))$.

Table 1: Average accuracy (%) across all phases using 7 state-of-art methods (span all sorts of CL) w/ and w/o C-Flat plugged in. *Maximum/Average Return* in the last row represents the maximum/average boost of C-Flat towards all methods in each column.

| Method | Technology | | | CIFAR-100 | | | ImageNet-100 | | Tiny-ImageNet |
| --- | --- | --- | --- | --- | --- | --- | --- | --- | --- |
| | Reg. | Mem. | Exp. | B0_Inc5 | B0_Inc10 | B0_Inc20 | B50_Inc10 | B50_Inc25 | B0_Inc40 |
| Replay [44] | | • | | 58.83 | 58.87 | 62.82 | **63.89** | 72.18 | 43.31 |
| w/ C-Flat | | | | **59.98** ↑ | **59.42** ↑ | **64.71** ↑ | 63.60 ↓ | **73.37** ↑ | **44.95** ↑ |
| iCaRL [43] | | • | | 58.66 | 59.76 | 61.13 | 64.78 | **77.25** | 45.70 |
| w/ C-Flat | | | | **59.13** ↑ | **60.40** ↑ | **62.93** ↑ | **65.01** ↑ | 76.22 ↓ | **46.08** ↑ |
| WA [64] | • | | | 63.36 | 66.76 | 68.04 | 73.17 | 80.81 | 55.69 |
| w/ C-Flat | | | | **65.70** ↑ | **67.79** ↑ | **69.16** ↑ | **73.56** ↑ | **83.84** ↑ | **56.06** ↑ |
| PODNet [11] | • | • | | 48.05 | 56.01 | 63.45 | 83.66 | 85.95 | 54.24 |
| w/ C-Flat | | | | **49.70** ↑ | **56.58** ↑ | **64.37** ↑ | **84.31** ↑ | **86.85** ↑ | **55.13** ↑ |
| DER [57] | | | • | 69.99 | 71.01 | 71.40 | 85.17 | 87.10 | 58.63 |
| w/ C-Flat | | | | **71.11** ↑ | **72.08** ↑ | **72.01** ↑ | **86.64** ↑ | **87.96** ↑ | **60.14** ↑ |
| FOSTER [54] | • | | • | 63.15 | 66.73 | 69.70 | 84.54 | 87.81 | 58.80 |
| w/ C-Flat | | | | **63.58** ↑ | **67.34** ↑ | **70.89** ↑ | **85.40** ↑ | 87.81 - | **58.88** ↑ |
| MEMO [68] | | | • | 67.42 | 69.82 | 69.91 | 67.28 | 83.09 | 58.15 |
| w/ C-Flat | | | | **67.56** ↑ | **69.94** ↑ | **71.79** ↑ | **69.34** ↑ | **83.41** ↑ | **58.97** ↑ |
| *Average Return* | | | | *+1.04%* | *+0.66%* | *+1.34%* | *+0.62%* | *+0.90%* | *+0.81%* |
| *Maximum Return* | | | | *+2.34%* | *+1.07%* | *+1.89%* | *+2.06%* | *+3.03%* | *+1.64%* |

Foster uses KL-based loss function to regularize the three combinations of old and new blocks for a stable performance on the previous data. It also introduces an effective redundant parameters and feature pruning strategy to maintain the single backbone model using knowledge distillation. DER follows the same framework, and introduces an auxiliary classifier and related loss item to encourage the model to learn diverse and discriminate features for novel concepts.

**Solution:** for expansion-based approaches, the plug-and-play strategy is still available. The C-Flat loss can be reformed with the reconstructed model as eq. 12. Thus C-Flat optimization using algorithm 1 is applied onto the first stage, where the new constructed block are optimized, while the generalized blocks are kept frozen. The final model is obtained after post-processing.

$$\ell^{C}_{\hat{S^T}}(f^T) = \ell^{R^0_\rho}_{\hat{S^T}}([\psi^{old}, \psi^{new}]([\phi_f^{T-1}(\phi_g), \phi_f^{new}(\phi_g)]))$$
$$+ \lambda \cdot \ell^{R^1_\rho}_{\hat{S^T}}([\psi^{old}, \psi^{new}]([\phi_f^{T-1}(\phi_g), \phi_f^{new}(\phi_g)])). \tag{12}$$

**To conclude**, C-Flat can be easily applied to any CL method with reconstructed loss function, and thus trained with the corresponding optimize as shown in algorithm 1. Dedicated design using C-Flat like for the GPM family is also possible wherever flat minima is required.

## 4 Analysis

### 4.1 Experimental Setup

**Datasets.** We evaluate the performance on CIFAR-100, ImageNet-100 and Tiny-ImageNet. Adherence to [66, 67], the random seed for class-order shuffling is fixed at 1993. Subsequently, we follow two typical class splits in CIL: (i) Divide all $\|Y_b\|$ classes equally into $B$ phases, denoted as **B0_Inc**$y$; (ii) Treat half of the total classes as initial phases, followed by an equally division of the remaining classes into incremental phases, denoted as **B50_Inc**$y$. In both settings, $y$ denotes that learns $y$ new classes per task.

**Baselines.** To evaluate the efficacy of our method, we plug it into 7 top-performing baselines across each CL category: Replay [44] and iCaRL [43] are classical replay-based methods, using raw data as memory cells. PODNet [11] is akin to iCaRL, incorporating knowledge distillation to constraint the logits of pooled representations. WA [64] corrects prediction bias via regularizing discrimination and fairness. DER [57], FOSTER [54] and MEMO [68] are network expansion

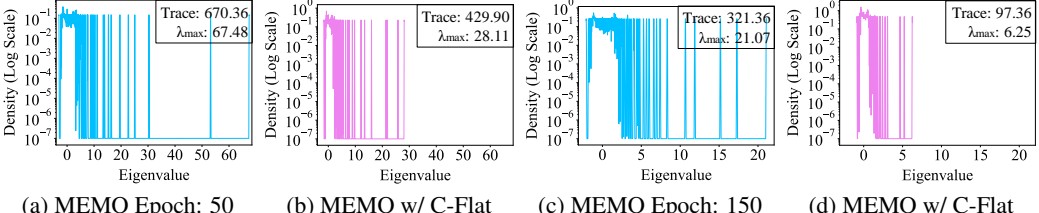

(a) MEMO Epoch: 50    (b) MEMO w/ C-Flat    (c) MEMO Epoch: 150    (d) MEMO w/ C-Flat

Figure 2: The Hessian eigenvalues and the traces at epochs 50, and 150 on B0_Inc10 setting (MEMO, CIFAR-100) w/ and w/o C-Flat plugged in.

methods, dedicate modular architectures towards each task by extending sub-network or freezing partial parameters. The aforementioned methods span three categories in CL [9, 53]: Memory-based methods, Regularization-based methods and Expansion-based methods.

**Network and training details.** For a given dataset, we study all methods using the same network architecture following repo [66, 67], *i.e.* ResNet-32 for CIFAR and ResNet-18 for ImageNet. If not specified otherwise, the hyper-parameters for all models adhere to the settings in the open-source library [66, 67]. Each task are initialized with the same $\rho$ and $\eta$, which drops with iterations according to the scheduler from [70]. To ensure a fair comparison, all models are trained with a vanilla-SGD optimizer [71]. And the proposed method is plugged into the SGD.

## 4.2 Make Continual Learning Stronger

Table 1 empirically demonstrates the superiority of our method: Makes Continual Learning Stronger. In this experiment, we plug C-Flat into 7 state-of-the-art methods that cover the full range of CL methods. From Table 1, we observe that (i) C-Flat presents consistent outperformance on all models, spanning Memory-based methods, Regularization-based methods, and Expansion-based methods. This superiority is indicative of the plug-and-play feature inherent in our method, allowing effortless installation with all sorts of CL paradigms. (ii) Across multiple benchmark datasets, including CIFAR-100, ImageNet-100, and Tiny-ImageNet, C-Flat exhibits consistent improvement. This underscores its generalization ability and effectiveness against diverse data distributions. (iii) C-Flat presents consistent boosting across multiple incremental scenarios, encompassing B0_Inc5, B0_Inc10, B0_Inc20, B50_Inc10, B50_Inc25, and B0_Inc40. This consistent boosting reaffirms robustness of C-Flat for various CL scenarios. To sum up, C-Flat advances baselines across each CL category, serves as a valuable addition to CL, offering a versatile solution that can complement existing methods.

## 4.3 Hessian Eigenvalues and Hessian Traces

**Hessian eigenvalues.** Equation 10 delineates the connection between fist-order flatness and Hessian eigenvalues in CL. Broadly, Hessian eigenvalues serve as a metric for assessing the flatness of a function. Thus we report Hessian eigenvalue distributions in Figure 2 for empirical analysis. As shown in Figure 2, models trained with vanilla-SGD exhibit higher maximal Hessian eigenvalues (67.48/21.07 at epochs 50/150 in Figure 2a and Figure 2c), while our method induces a significant drop in Hessian eigenvalues to 28.11/6.25 at epochs 50/150 in Figure 2b and Figure 2d) during CL, leading to flatter minima. Consequently, the performance of CL is tangibly enhanced.

**Hessian traces.** We calculate the empirical Fisher information matrix as an estimation of the Hessian and leverage the trace of this to quantify the flatness of the approximation loss at the convergence point. As depicted in Figure 2, we observe that a substantial reduction in the Hessian trace when employing our method compared with vanilla-SGD (670.36/321.36 drops to 429.90/97.36 at epochs 50/150 in Figure 2b and Figure 2d). This observation suggests that our method induces a flatter minimum. These findings not only align with but also substantiate the theoretical insights presented in the methodology section.

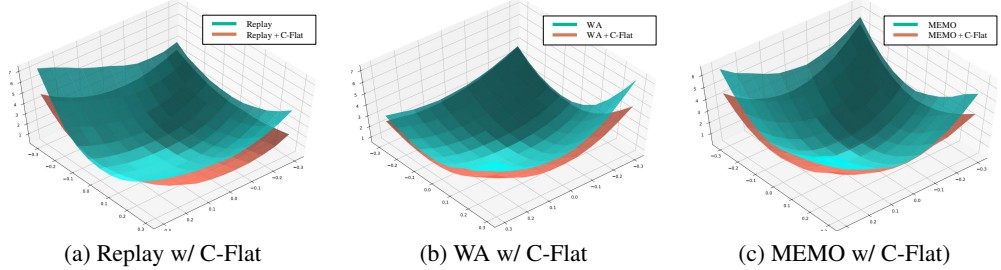

| (a) Replay w/ C-Flat | (b) WA w/ C-Flat | (c) MEMO w/ C-Flat) |

Figure 3: The parametric loss landscapes of Replay (Mem.), WA (Reg.) and MEMO (Exp.) are plotted by perturbing the model parameters at the end of training (CIFAR-100, B0_Inc10) across the first two Hessian eigenvectors.

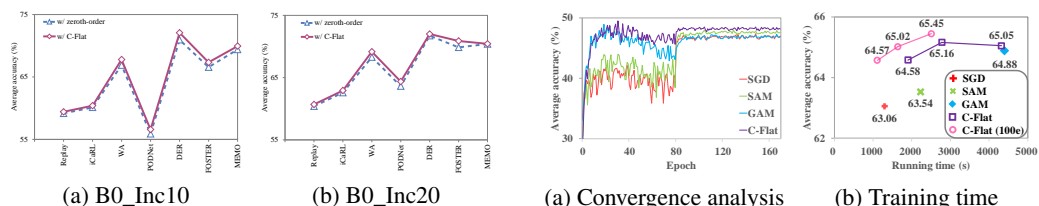

| (a) B0_Inc10 | (b) B0_Inc20 | | (a) Convergence analysis | (b) Training time |

Figure 4: C-Flat vs. Zero-order flatness

Figure 5: Analysis of computation overhead

## 4.4 Visualization of Landscapes

More intuitively, we present a detailed visualization of landscape. PyHessian [59] is used to draw the loss landscape of models. To simplify, we choose one typical method from each category of CL methods (Replay, Wa, MEMO) for testing. Figure 3 clearly illustrates that, by applying C-Flat, the loss landscape becomes much flatter than that of the vanilla method. This trend consistently holds across various categories of CL methods, providing strong empirical support for C-Flat, and confirms our intuition.

## 4.5 Revisiting Zeroth-order Flatness

Table 2: Revisiting FS-DGPM series using C-Flat.

| Method | La-GPM | FS-GPM | DGPM | La-DGPM | FS-DGPM |
|--------|--------|--------|------|---------|---------|
| Oracle | 72.90 | 73.12 | 72.66 | 72.85 | 73.14 |
| w/ C-Flat | **73.66** | **73.57** | **73.01** | **73.64** | **73.72** |
| *Boost* | **+0.76** | **+0.45** | **+0.35** | **+0.79** | **+0.58** |

Limited work [10, 49] proved that the zeroth-order sharpness leads to flat minima boosted CL. Here, we employ a zeroth-order optimizer [19] instead of vanilla-SGD to verify the performance of C-Flat. As shown in Figure 4, C-Flat (purple line) stably outperforms the zeroth-order sharpness (blue line) on all baselines. We empirically demonstrated that flatter is better for CL.

Former work FS-DGPM [10] regulates the gradient direction with flat minima to promote CL. The FS (Flattening Sharpness) term derived from FS-DGPM is a typical zeroth-order flatness. We revisit the FS-DGPM series (including La/FS-GPM, DGPM, La/FS-DGPM) [10, 46] to evaluate performance using C-Flat instead of FS (see aigorithm 2). Table 2 yields two conclusions: (i) C-Flat boosts the GPM [46] baseline as a pluggable regularization term. This not only extends the frontiers of CL methods, incorporating gradient-based solutions, but also reaffirms the remarkable versatility of C-Flat. (ii) Throughout all series of FS-DGPM, C-Flat seamlessly supersedes FS and achieves significantly better performance. This indicates that C-Flat consistently exceeds zeroth-order sharpness. Hence, reconfirming that C-Flat is indeed a simple yet potential CL method that deserves to be widely spread within the CL community.

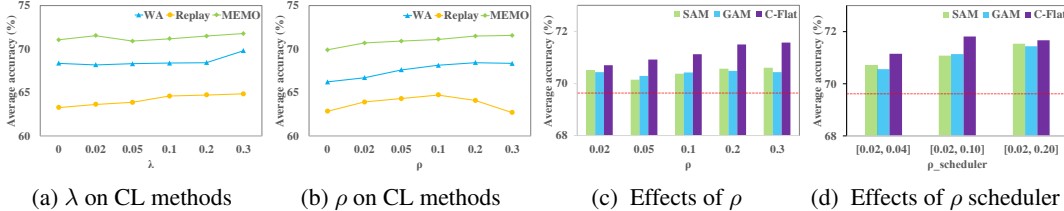

| (a) $\lambda$ on CL methods | (b) $\rho$ on CL methods | (c) Effects of $\rho$ | (d) Effects of $\rho$ scheduler |

Figure 6: Ablation study about $\lambda$ and $\rho$. (a) and (b) represents the effect of $\lambda$ and $\rho$ on different CL methods (WA, Replay, MEMO). (c) and (d) represents the effect of $\rho$ and $\rho$ scheduler on MEMO with different optimizers (SGD (red line), SAM, GAM, C-Flat).

## 4.6 Computation Overhead

To assess the efficiency of C-Flat, we provides a thorough analysis from the convergence speed and running time with CIFAR-100/B0_Inc20 on Replay. As shown in Figure 5, C-Flat is compared with SGD and other flatness-aware optimiziters. We train C-Flat optimizers on CL benchmarks with 20%, 50%, 100% of iterations and approximately 60% of epochs, while holding the other optimizers at 100%. Figure 5a first shows that C-Flat converges fastest and has the highest accuracy (purple line), meaning few iterations/epochs with C-Flat is enough to improve CL. Figure 5b shows i) Compared with SGD, with only 20% of iterations and 60% of epochs (pink line) using C-Flat, CL performance is improved using slightly less time; ii) C-Flat surpasses GAM with similar time as SAM when setting the iterations/epochs ratio to 50%/60%; iii) Models trained with C-Flat for 100 epochs outperform those trained with other optimizers for 170 epochs. To sum up, we show that C-Flat outperforms current optimizers with fewer iterations and epochs. This indicates the efficiency of C-Flat.

To discuss practicality better, we provided a tier guideline, which categorizes C-Flat into L1 to L3 levels, as shown in Table 3, L1 denotes the low-speed version of C-Flat, with a slightly lower speed than SAM and the best performance; L2 follows next; L3 denotes the high-speed version of C-Flat, with a faster speed than SGD and a performance close to L2.

Table 3: A tier guideline of C-Flat.

| Level | Speed | Boost (SGD/SAM) |
| --- | --- | --- |
| L1 | SGD > SAM > **C-Flat** | +2.39%/+1.91% |
| L2 | SGD > **C-Flat** > SAM | +1.52%/+1.04% |
| L3 | **C-Flat** > SGD | +1.51%/+1.03% |

## 4.7 Ablation Study

We perform ablation study in two cases: (i) the influence of $\lambda$ and $\rho$ on different CL methods; (ii) the influence of $\rho$ and its scheduler on different optimizers.

We first present the performance of C-Flat with varying $\lambda$ and $\rho$. As described in Eq. 13, $\lambda$ controls the strength of the C-Flat penalty (when $\lambda$ is equal to 0, this means that first-order flatness is not implemented). As shown in Figure 6a, compared with vanilla optimizer, C-Flat shows remarkable improvement with varying $\lambda$. Moreover, $\rho$ controls the step length of gradient ascent. As shown in Figure 6b, C-Flat with $\rho$ larger than 0 outperforms C-Flat without gradient ascent, showing that C-Flat benefits from the gradient ascent.

For each CL task $T$, same learning rate $\eta^T$ and neighborhood size $\rho^T$ initialization are used. By default, $\rho_i^T \in [\rho_-, \rho_+]$ is set as a constant, which decays with respect to the learning rate $\eta_i^T \in [\eta_-, \eta_+]$ by $\rho_i^T = \rho_- + \frac{(\rho_+ - \rho_-)}{\eta_+ - \eta_-}(\eta_i^T - \eta_-)$. Figure 6c and Figure 6d present a comparison on $\rho$ initialization and $\{\rho_-, \rho_+\}$ scheduler. C-Flat outperforms across various settings, and is not oversensitive to hyperparameters in a reasonable range.

## 4.8 Beyond Not-forgetting

As is known to all, forward, and in particular backward transfer, are the desirable conditions for CL [22]. Here, we thoroughly examine the performance of C-Flat in both aspects. Forward Transfer (FT) means better performance on each subsequent task. Backward Transfer (BT) means better

Figure 7: Analysis of BT and FT. RR refers to Relative Return on w/o and w/ C-Flat.

| Method | | CIFAR-100/ B0_Inc5 | | |
| --- | --- | --- | --- | --- |
| | | w/o C-Flat | w/ C-Flat | RR |
| iCaRL [43] | old | 36.36 | 37.12 | BT+2.10% |
| | new | 80.25 | 82.20 | FT+2.43% |
| PODNet [11] | old | 46.32 | 47.44 | BT+2.42% |
| | new | 62.65 | 64.75 | FT+3.35% |
| FOSTER [54] | old | 58.50 | 61.35 | BT+2.85% |
| | new | 62.05 | 63.05 | FT+1.61% |

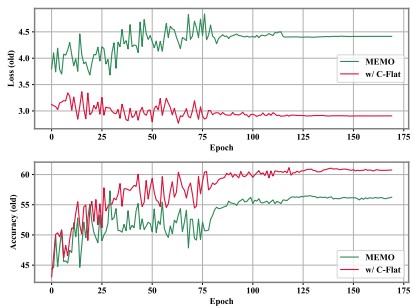

Figure 8: Loss and forgetting of old tasks.

performance on previous tasks, when revisited. We count the performance of new and old tasks on several CL benchmarks before and after using C-Flat. As observed in Table 7, C-Flat consistently improves the learning performance of both new and old tasks. This observation indicates that C-Flat empowers these baselines with robust forward and backward transfer capabilities, that is learning a task should improve related tasks, both past and future. But, thus far, achieving a baseline that maintains perfect recall (by forgetting nothing) remains elusive. Should such a baseline emerge, C-Flat stands poised to empower it with potent backward transfer, potentially transcending the limitations of mere not-forgetting.

Moreover, one of our contributions is to prove the positive effect of low curvature on overcoming forgetting. Intuitively, we visualized the change in loss and forgetting of old tasks in CL. Figure 8 shows the lower loss or less forgetting (red line) for old tasks during CL. This is an enlightening finding.

## 5 Conclusion

This paper presents a versatile optimization framework, C-Flat, to confront forgetting. Empirical results demonstrate C-Flat's consistently outperform on all sorts of CL methods, showcasing its plug-and-play feature. Moreover, the exploration of Hessian eigenvalues and traces reaffirms the efficacy of C-Flat in inducing flatter minima to enhance CL. In essence, C-Flat emerges as a simple yet powerful addition to the CL toolkit, making continual learning stronger.

## 6 Acknowledgments

This work was supported in part by the Chunhui Cooperative Research Project from the Ministry of Education of China under Grand HZKY20220560, in part by the National Natural Science Foundation of China under Grant W2433165, and in part by the National Natural Science Foundation of Sichuan Province under Grant 2023YFWZ0009.

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

# A Appendix

## A.1 Overview

In this supplementary material, we first present more intuitive visualizations of C-Flat, elucidating the loss landscape from local viewpoint (Appendix A.5.1) and each task during CL (Appendix A.5.2). Next, we provide more details on the accuracy and runtime trade-offs of other CL methods with our C-Flat (Appendix A.6)

## A.2 C-Flat

We summarize the pseudo code of C-Flat in algorithm 1.

---

**Algorithm 1** C-Flat Optimization

---

**Input:** Training phase $T$, training data $S^T$, model $f^{T-1}$ with parameter $\theta^{T-1}$ from last phase if $T > 1$, batch size $b$, oracle loss function $\ell$, learning rate $\eta > 0$, neighborhood size $\rho$, trade-off coefficient $\lambda$, small constant $\epsilon$.

**Output:** Model trained at the current time $T$ with C-Flat.

**Initialization:**

**if** T=1: **then** Randomly Initialize parameter $\theta^{T=1}, \eta^{T=1} = \eta, \rho^{T=1} = \rho$.

**else**

  Reconstruct the model and training set if necessary,

  Initialize model parameter $\theta^T$ according to the training strategy, like randomly initialization or $\theta^T = \theta^{T-1}$ in pre-trained model based approaches, $\eta^T = \eta, \rho^T = \rho$,

  Frozen part of the parameter if required.

**end if**

**Optimization:**

**while** $\theta^T$ not converge, **do**

  Sample batch $B^T$ of $b$ random instances from $S^T$

  Compute batch's loss gradient $g_{B^T} = \bigtriangledown \ell_{B^T}(f^T(\theta^T))$

  Compute $R_\rho^0$ perturbation: $\epsilon_0 = \rho^T \cdot \frac{|g_{B^T}|}{\|g_{B^T}\|_2 + \epsilon}$

  Approximate zeroth-order gradient: $g_0 = \bigtriangledown \ell_{B^T}(f^T(\theta^T + \epsilon_0))$

  Compute hessian vector product: $h_{B^T} = \bigtriangledown^2 \ell_{B^T}(\theta) \cdot \frac{\bigtriangledown \ell_{B^T}(f^T(\theta^T))}{\|\bigtriangledown \ell_{B^T}(f^T(\theta^T))\|_2 + \epsilon}$

  Compute $R_\rho^1$ perturbation: $\epsilon_1 = \rho^T \cdot \frac{h_{B^T}}{\|h_{B^T}\|_2 + \epsilon}$

  Approximate first-order gradient: $g_1 = \bigtriangledown^2 \ell_{B^T}(f^T(\theta^T + \epsilon_1)) \cdot \frac{\bigtriangledown \ell_{B^T}(f^T(\theta^T + \epsilon_1))}{\|\bigtriangledown \ell_{B^T}(f^T(\theta^T + \epsilon_1))\|_2 + \epsilon}$

  Update: Model parameter: $\theta^T = \theta^T - \eta^T(g_0 + \lambda g_1)$; Update training parameters $\eta^T, \rho^T$ according to a scheduler that the values drop with iterations;

**end while**

**Post-Processing** on Model and Training data if required.

  **return** Model $f^T$ with parameter $\theta^T$

---

## A.3 C-Flat-GPM

We summarize the pseudo code of C-Flat for GPM family in algorithm 2.

## A.4 Convergency Proof

**Assumptions 1:** the loss function is twice differentiable, bounded by $M$, with bounded variance $\sigma^2$, and obeys the triangle inequality. Both the loss function and its second-order gradient are $\beta-$Lipschitz smooth. $\eta \leq 1/\beta, \rho \leq 1/4\beta$, and $\eta_i^T = \eta/\sqrt{i}, \rho_i^T = \rho/\sqrt[4]{i}$ for epoch $i$ in any task $T$.

**Algorithm 2** C-Flat for GPM-family at $T > 1$

---

**Input:** Training set $\hat{S^T}$, parameter $\theta^T = \theta^{T-1}$, loss $\ell$, learning rate $\eta_1, \eta_2$, basis matrix $\mathbb{M}$ and significance $\Lambda$ from replay buffer.
**while** $\theta^T$ not converge, **do**
    Sample batch $B^T$
    Compute perturbation $\epsilon_c$ using C-Flat optimization
    Update basis significance: $\Lambda = \Lambda - \eta_1 \cdot \nabla_\Lambda \ell_{B^T}(\theta^T + \epsilon_c)$
    Update model parameter: $\theta^T = \theta^T - \eta_2 \cdot (I - \mathbb{M}\Lambda M) \bigtriangledown \ell_{B^T}(\theta^T + \epsilon_c)$
    Update $M$ and replay buffer.
**end while**
    **return** Model parameter $\theta^T$

---

**Claim 1:** with **Assumptions 1**, the convergency of zeroth-sharpness with batch size $b$ is guaranteed [3] by

$$\frac{1}{n}\sum_{i=1}^{n}\mathbb{E}[\|\nabla\ell^{R_\rho^0}\|^2] \leq \frac{4\beta}{\sqrt{n^T}}[\ell(\theta) - \ell(\theta^*)] + \frac{8\sigma^2}{3b\sqrt{n^T}} \tag{13}$$

hence, the zeroth-order part of C-Flat is bounded:

$$\frac{1}{n^T}\sum_{i=1}^{n^T}\mathbb{E}[\|\nabla\ell_{S^T}^{R_\rho^0}(f^T(\theta^T))\|^2] \leq \frac{4\beta}{\sqrt{n^T}}[\ell_{S^T}(f^T(\theta^T))] + \frac{8\sigma^2}{3b\sqrt{n^T}} \leq \frac{4M\beta}{\sqrt{n^T}} + \frac{8\sigma^2}{3b\sqrt{n^T}} \tag{14}$$

**Lemma 1:** let $\xi_{tr}(\theta) = \ell_{tr}(f^T(\theta), f^T(\theta^*))$, with **Assumptions 1**, the first-order part is bounded by

$$\frac{1}{n^T}\sum_{i=1}^{n^T}\mathbb{E}[\|\nabla\ell_{S^T}^{R_\rho^1}(f^T(\theta^T))\|^2] \leq \frac{1}{n^T}\sum_{i=1}^{n^T}\mathbb{E}[\|\nabla\xi_{tr}(\theta^T + \epsilon_1) - \nabla\xi_{tr}(\theta^T))\|^2]$$

$$\leq\frac{\beta^2}{n^T}\sum_{i=1}^{n^T}\mathbb{E}[\|\epsilon_1\|^2] \leq \frac{\beta^2\eta^2}{n^T}\sum_{i=1}^{n^T}\mathbb{E}[\|\rho_i^T\|^2] \leq \frac{\rho^2}{n^T}\sum_{i=1}^{n^T}\mathbb{E}[i^{-2}] \leq \frac{16(2\sqrt{n^T}-1)}{\beta^2 n^T} \tag{15}$$

**Theorem 1:** with **Assumptions 1**, by combining the zeroth- and first-order parts, we can prove C-Flat converges in all tasks that,

$$\frac{1}{n^T}\sum_{i=1}^{n^T}\mathbb{E}[\|\nabla\ell_{S^T}^C(f^T(\theta^T))\|^2] \leq \frac{2}{n^T}\sum_{i=1}^{n^T}\mathbb{E}[\|\nabla\ell_{S^T}^{R_\rho^0}(f^T(\theta^T))\|^2]$$

$$+\frac{2}{n^T}\sum_{i=1}^{n^T}\mathbb{E}[\|\lambda R_{\rho,S^T}^1(f^T(\theta^T))\|^2] \leq \frac{8M\beta}{\sqrt{n^T}} + \frac{16\sigma^2}{3b\sqrt{n^T}} + \frac{32\lambda^2(2\sqrt{n^T}-1)}{\beta^2 n^T}. \tag{16}$$

### A.5 More Visualizations of C-Flat

In this section, we present additional visualization of the loss landscape involving two cases using PyHessian [59]: (i) Changes in the loss landscape from localized viewpoints; (ii) Changes in the loss landscape across each task during CL.

### A.5.1 Landscapes in a Local Viewpoint

First, we present a more detailed visualization through changes in the local region of the loss landscape. We set a minimal radius threshold. At this threshold, more detailed changes are displayed. Similarly, we choose three typical method (Replay [44], WA [64], MEMO [68]) from each category of CL methods for visualization. As shown in Fig. 9, in a tiny view, C-Flat contributes to a flatter surface, a change that more intuitively reveals the mechanism of C-Flat.

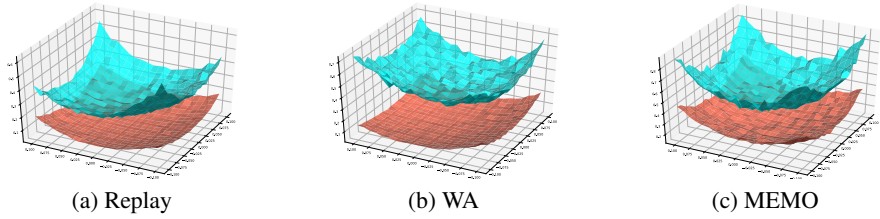

(a) Replay        (b) WA        (c) MEMO

Figure 9: The visualizations of loss landscapes in a local viewpoint (Replay, WA and MEMO)

### A.5.2 Landscapes Across Each Task

Second, we further visualize the loss landscape of more tasks using PyHessian for more intuitive explanations during CL. To simplify, we choose one CL method (Replay [44]) for visualization on task 2, 7, 12 and 17 with 5 task intervals. As shown in Fig. 10 (a) to (d), the loss landscape all becomes much flatter than that of the vanilla method across each task. This trend provides strong empirical support for C-Flat.

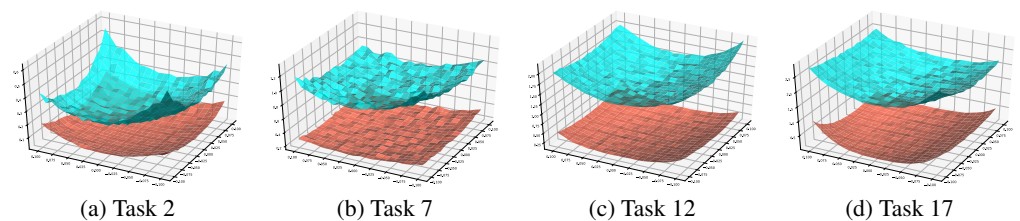

(a) Task 2     (b) Task 7     (c) Task 12     (d) Task 17

Figure 10: The visualizations of loss landscapes during CL.

### A.6 Overhead of C-Flat

To enhance the computing efficiency, we apply C-Flat in a limited number of iterations within each epoch. Remarkably, we observe that without executing C-Flat in every iteration can also significantly boost the performance of CL (All cases derived from C-Flat improves CL performance). As illustrated in Table 4, 10% C-Flat iterations is enough to improve CL performances, and around 50% C-Flat iterations is enough to approach and even exceed the impact of a full C-Flat training. As a consequence, the overhead of 50% C-Flat is at least 30% shorter compared with the full C-Flat training. These observations holds potential for light C-Flat boosted CL applications.

Table 4: Accuracy and training speed of training with different ratios of iterations using C-Flat. Superscripts denotes the ratio of iterations in each epoch is trained with 100%, 50%, 20% and 10%.

| Method | C-Flat$^1$ | C-Flat$^{0.5}$ | C-Flat$^{0.2}$ | C-Flat$^{0.1}$ | Oracle |
|---|---|---|---|---|---|
| Replay [44] | 61.02 **(100%)** | 60.98 **(67%)** | 60.63 **(40%)** | 60.48 **(34%)** | 60.28 |
| iCaRL [43] | 63.04 **(100%)** | 62.94 **(65%)** | 62.78 **(41%)** | 62.75 **(35%)** | 62.74 |
| WA [64] | 68.67 **(100%)** | 68.20 **(59%)** | 67.96 **(38%)** | 68.02 **(31%)** | 67.75 |
| PODNet [11] | 64.35 **(100%)** | 63.82 **(60%)** | 63.27 **(39%)** | 63.80 **(34%)** | 63.05 |
| DER [57] | 72.25 **(100%)** | 71.82 **(59%)** | 71.82 **(39%)** | 71.44 **(31%)** | 71.52 |
| FOSTER [54] | 70.24 **(100%)** | 70.43 **(70%)** | 69.99 **(52%)** | 69.71 **(47%)** | 69.30 |
| MEMO [68] | 69.97 **(100%)** | 70.03 **(64%)** | 70.48 **(41%)** | 69.90 **(32%)** | 69.71 |

