# OpenReview forum: "Make Continual Learning Stronger via C-Flat"
_NeurIPS.cc/2024/Conference — NeurIPS 2024 poster_

### Official Review · Reviewer_475M · 2024-07-06

**Soundness:** 3
**Presentation:** 3
**Contribution:** 3
**Rating:** 8
**Confidence:** 4

**Summary:**

This paper points out that the current sharpness in continual learning (CL) tends to optimize towards a suboptimal space rather than achieving a global solution for continuous tasks. The proposed C-Flat, smoothly migrates to the global optimal of the joint knowledge space of the current and next tasks, thereby promoting CL. The idea is an intriguing one and has the potential to make a significant contribution to the CL community.

**Strengths:**

1) the paper is well-structured and presented, compelling narrative. The benefits and motivation generally flowed well. To their credit, the authors provide a theoretical analysis.

2) This paper investigates the relationship between flatness, loss landscape, and CL from an optimization view, and studies the limitation of sharpness applied in current works (e.g., FS-DGPM/F2M etc.), effectively complementing the current body of research.

3) The authors unify the three preceding aspects, and demonstrate the mechanism of loss landscape flatness on several categories of CL methods. The authors show the necessity of a flatter loss landscape for CL cuz favors global optimal during CL. Then, their optimization framework can be plugged into any CL method. Such an approach seems to be non-trivial. Third, this work may be a basic toolkit for the CL community.

**Weaknesses:**

1) The authors seem made a non-trivial effort in the exploration of the sharpness-aware approach in CL. However, they should elaborate more on the differences with the current CL methods in INTRODUCTION to highlight the contribution of the manuscript, although I note that they did so in RELATED WORK.

2) In Equation 3 and Equation 4, I noticed that C-Flat additionally decreases the curvature around local minima. Does this mean lower loss or less forgetting for old tasks in the next stage?

3) Their method shows an outstanding convergence speed (Figure 5), thus, in practice, the overhead is acceptable (close to SGD, SAM in some cases). Perhaps the authors could put more emphasis on practicality in SECTION 4.6, which could be meaningful.

4) A few terms (e.g., SAM, Figure 1, etc.) lack citations, which may hinder reading.

**Questions:**

All experiments are conducted on the complicated Class IL scenarios and consistently achieve improvements. However, still wondering how much the larger domain gap affects the stability of C-Flat. Although Class IL already has fairly complex data distribution changes for CL.

In addition, please answer WEAKNESS i, and ii.

**Limitations:**

Please see my comments in WEAKNESS and QUESTIONS.

---

> ### Author Rebuttal · Authors · 2024-08-06
>
> > Weakness 1: Improve INTRODUCTION.
>
> A: We have done this for the Introduction section to highlight contributions, including highlighting our CL-friendly optimizers, and also discussing how flatness-related works in current CL (flatness in some data/bases/projection, etc).
>
> > Weakness 2: The relationship between low curvature and less forgetting.
>
> A: Exactly, one of our contributions is to prove the positive effect of low curvature on overcoming forgetting. Intuitively, we visualized the change in loss and forgetting of old tasks during CL. Figure R1 in one page shows the lower loss or less forgetting (red line) for old tasks during CL. This is an enlightening finding.
>
> > Weakness 3: Discuss the practicality.
>
> A: To discuss practicality better, we provided a tier guideline, which categorizes C-Flat into L1 to L3 levels, as shown below,
>
> | Level | Version | Speed                           | Performance (SGD/SAM) |
> |:-----:|:-------:|:-------------------------------:|:---------------------:|
> |  L1   |   Low   |    SGD > SAM > **C-Flat**       |    +2.39% / +1.91%    |
> |  L2   | Medium  |    SGD > **C-Flat** > SAM       |    +1.52% / +1.04%    |
> |  L3   |  High   |      **C-Flat** > SGD           |    +1.51% / +1.03%    |
>
> As the table above shows, L1 denotes the low-speed version of C-Flat, with a slightly lower speed than SAM and the best performance; L2 follows next; L3 denotes the high-speed version of C-Flat, with a faster speed than SGD and a performance close to L2. We have updated this in the revision.
>
> > Weakness 4: Fix misc.
>
> A: We have fixed them in our manuscript, including citations, minor typos, etc.
>
> > Question 1: The robustness of C-Flat on larger domain gap.
>
> A: As you mentioned, the performance in the Class IL scenario has already demonstrated the robustness of C-Flat given the fairly complex domain shifts. However, we would still be happy to conduct a further test on C-Flat to ensure its reliability. As shown in Table below, we trained on ImageNet21K and then evaluated CL performance on IN-R/ObjNet. Unlike CIFAR, both datasets are acknowledged to have a large domain gap with ImageNet. The Table below shows that C-Flat maintains stability in more difficult cases.
>
> |   Method   | IN-R B0_Inc20   | ObjNet B0_Inc10  |
> |:----------:|:---------------:|:---------------:|
> |   iCaRL    |      72.13      |      48.06      |
> | w/ C-Flat  |  **72.92** (+0.79)  |  **49.59** (+1.53)  |
> |   MEMO     |      70.96      |      56.22      |
> | w/ C-Flat  |  **71.69** (+0.73)  |  **56.50** (+0.28)  |

---

> > ### Comment · Reviewer_475M · 2024-08-11
> >
> > Throughout the author's feedback and other reviewers' comments, I lean towards acceptance due to the non-trivial efforts and well-supported theory of this work for CL.
> >
> > In their rebuttal, my concern about the effect of low curvature on forgetting is adequately addressed and this offered new findings for CL. The performance on a larger domain gap also makes this work more solid.
> > Moreover, the reviewers all agree that this work is versatile across a variety of CL methods. The authors provide detailed evidence and details to address concerns raised by other reviewers. Reviewer Tb4q also recognized the valuable and in-depth theoretical analysis of this work. With my experience in CL, I would expect to use this optimizer in our future work after their codes are released.
> >
> > Overall, I still recommend this work to be accepted. Finally, I would like to request the authors to release their code if accepted.

---

> ### Author Response · Authors · 2024-08-11
>
> Thank you for your valuable comments and for acknowledging our work. We commit to releasing the code to the CL community.
>
> We appreciate your time and effort in reviewing our work! Thanks!

---

### Official Review · Reviewer_Tb4q · 2024-07-08

**Soundness:** 4
**Presentation:** 3
**Contribution:** 4
**Rating:** 6
**Confidence:** 4

**Summary:**

This paper proposes a novel method named C-Flat to mitigate catastrophic forgetting by optimizing for a flatter loss landscape. This method is described as a plug-and-play solution that can be easily integrated into a wide range of existing CL approaches. The paper argues that this approach not only stabilizes the learning process but also enhances the model's generalization across tasks by enabling it to find and utilize flatter minima. The authors demonstrate the effectiveness of their method when applied to different CL solutions on 3 different datasets.

**Strengths:**

1. **Novelty and Effectiveness of C-Flat**: The introduction of C-Flat as a method that emphasizes flatness in the loss landscape to address catastrophic forgetting is innovative. The paper provides extensive experimental results showing that C-Flat improves performance across a variety of continual learning settings and benchmarks.

2. **Theoretical Analysis**: The theoretical grounding provided for the convergence of the proposed loss is valuable and adds depth to the paper.

3. **Visualizations**: The visualization of the loss landscape in Figure 4 effectively illustrates the impact of the proposed method, providing a clear comparative insight into how C-Flat modifies the learning dynamics.

**Weaknesses:**

1. **Generalization Claims Overstated**: The paper's assertion that it resolves the sensitivity/stability dilemma in CL is overly strong. While the proposed method shows promising results, it would be more accurate to state that it addresses rather than resolves these issues. (see line 96)

2. **Theoretical Section Structure**: The structure of the theoretical analysis section is confusing. It would benefit from a more organized presentation, starting with assumptions, followed by theorems, and a detailed proof. This would enhance the readability and academic rigor of the paper. The assumptions (such as the differentiability of the loss) should be stated first in an assumption environment. Then the theorem or proposition should be stated (for example: Given the assumption above, when <conditions>, the convergence of <loss> is guaranteed) followed by a proof environment. clumping several steps in one equation (Eq. 8) is confusing and should be avoided for clarity.

3.**Oragnization**: From the beginning of line 141 to the end of Eq. 10 seems disconnected from the rest of the paper. The authors should provide context for what they are proposing and explain why the proposed content matters.

**Questions:**

1. **Revision of Language**: The authors are encouraged to thoroughly revise the language of the manuscript to enhance readability and clarity.

2. **Typo**: I believe there is a typo in line 98.

3. **Average Boost**: I would like to ask the authors to report the average of the boost alongside the maximum boost in the results section. This could give a better understanding of the effectiveness of the proposed method.

**Limitations:**

The paper lacks a discrete limitation section or a discussion of the limitations in the conclusion section. I would like to ask the authors to include this in their paper.

---

> ### Author Rebuttal · Authors · 2024-08-06
>
> > Weakness 1: Overly strong claims.
>
> A: We appreciate your suggestion! We have revised the manuscript in line 96 to temper these statements.
>
> > Weakness 2: More clear structure for the method.
>
> A: We did shorten that part a lot into one paragraph than a better math environment due to the page limit. We have rewritten that part with highlighted 'assumption' and 'condition' and attached a detailed proof in the appendix. Many thanks for the patient explanation.
>
> > Weakness 3: Highlight the context of Eq.10
>
> A: Thanks for pointing it out. We intended to show some mathematical properties, i.e. the connection of C-Flat to Hessian matrix. Indeed, it does look disconnected. We have added a highlighted "Upper Bound" at the beginning of line 141 and some context about this property at the end of line 144 to make it clear.
>
> > Question 1: Improve readability and clarity.
>
> A: We have polished the manuscript again to improve readability and clarity, and fixed minor errors, like errors in line 38,40 and 98.
>
> > Question 2: Report average boost.
>
> A: We have updated maximum boost/average boost in Table 1 of the main paper as below,
>
> | Gains         | B0_Inc5 | B0_Inc10 | B0_Inc20 | B50_Inc10 | B50_Inc25 | B0_Inc40 |
> |---------------|---------|----------|----------|-----------|-----------|----------|
> | Maximum Boost | +2.34%  | +1.07%    | +1.89%   | +2.06%    | +3.03%    | +1.64%   |
> | Average Boost | +1.04%  | +0.66%   | +1.34%   | +0.62%    | +0.90%    | +0.81%   |
>
> > Limitation 1: Discuss limitations.
>
> A: C-Flat potentially has the following limitations. First, all experiments have not yet been validated on pre-training model (PTM)-based CL method. In the era of PTM, exploring the collaborative mechanisms between the proposed method and PTM is essential for advancing CL, which can be a promising future work. Second, C-Flat is under-explored in the transformer-based model, and this architecture is prevalent in the foundation model, which is remaining for future work. As above, we will update this discussion in the revision.

---

> > ### Comment · Reviewer_Tb4q · 2024-08-11
> >
> > Thank you to the authors for the additional reported results and the detailed response. I believe these changes can enhance the clarity of the paper. Due to the level of novelty in this work, I will be keeping my score the same.

---

> > > ### Author Response · Authors · 2024-08-11
> > >
> > > Thank you for recognizing our work. Your valuable suggestions significantly improved the quality of our manuscripts. We appreciate your time and effort in reviewing our work! Thanks a lot!

---

### Official Review · Reviewer_PuVp · 2024-07-09

**Soundness:** 2
**Presentation:** 3
**Contribution:** 3
**Rating:** 5
**Confidence:** 4

**Summary:**

Continual learning seeks to learn a series of new tasks without forgetting old ones. This paper explores the impact of a flat loss landscape on catastrophic forgetting.

**Strengths:**

- This paper applies loss landscape optimization to multiple categories of continual learning methods.
- The method proposed in this paper is easy to implement and the article structure is well organized.

**Weaknesses:**

- The contribution of this paper is unclear. The authors claim that they are the first to compare CL methods with loss landscapes. However, there are many works that have discussed the impact of flatness on continual learning (or catastrophic forgetting) [1-5], which are not discussed.
- The discussion of related work is not comprehensive. First, the connection and difference between the proposed method and the existing loss landscape-based continual learning methods [1-5] are not fully discussed in related work. In addition, there is also a lack of discussion on the difference and connection between the proposed C-Flat and various existing sharpness-aware minimization methods.
- There is a lack of performance comparison with related work, so it is unclear how the gains compare to other works that improve CL based on flatness [1, 2, 3].
- The proposed C-Flat has significantly larger computational cost. As mentioned in Section 3, it requires 2 forward and 4 backward propagations.
- The results in Tab.2 show that C-Flat has only a slight performance improvement.

References:

[1] Cha, S., Hsu, H., Hwang, T., Calmon, F. P., & Moon, T. CPR: classifier-projection regularization for continual learning. In ICLR, 2021.

[2] Yang, E., Shen, L., Wang, Z., Liu, S., Guo, G., & Wang, X.. Data augmented flatness-aware gradient projection for continual learning. In ICCV, 2023.

[3] Tran Tung, L., Nguyen Van, V., Nguyen Hoang, P., & Than, K.. Sharpness and gradient aware minimization for memory-based continual learning. In SOICT, 2023.

[4] Chen, Runhang, et al. "Sharpness-aware gradient guidance for few-shot class-incremental learning." Knowledge-Based Systems (2024): 112030.

[5] Mehta, S. V., Patil, D., Chandar, S., & Strubell, E. (2023). An empirical investigation of the role of pre-training in lifelong learning. Journal of Machine Learning Research, 24(214), 1-50.

**Questions:**

- In Section 2, why should “Gradient-based solutions” be a separate paragraph?
- In Figure 5, why does the loss become smooth after the 80th epoch, but it is very drastic before?

**Limitations:**

See Weaknesses

---

> ### Author Rebuttal · Authors · 2024-08-06
>
> **NOTE**: Table R1-R4 and Figure R1 are attached to the one-page PDF.
> > Weakness 1: Clarification of contributions.
>
> A: For the suggested works, they could further validate the importance of C-Flat on a general and stronger flatness-aware CL optimizer, we have cited them and the detailed discussion can be seen in our answer to Weakness 2.
>
> However, all these works are focused on the zeroth order sharpness, i.e. SAM for CL [5], and the improved works are also designed for specific kind of CL approaches like GPM method [2], memory based method [3] or certain CL scenario [4], while wide local minima in [1] is another concept about the model parameter's similarity to the last task.
>
> As a comparison, 1) we proposed a CL optimizer beyond zeroth order sharpness; 2) C-Flat can be generally applied to any CL methods including the aforementioned ones; 3) as can be seen in our reply to Weakness 4, thanks to the faster convergence speed, we still achieve optimal result with a few overhead compared with other SAM optimizer by performing C-Flat in periodical iterations.
>
> Moreover, for clear contributions, you also can refer to the Strengths 1 of **Reviewer p3Vh**; Strengths 1,2 of **Reviewer Tb4q** and Strengths 1,2 of **Reviewer 475M**.
>
> >Weakness 2: Improve related work.
>
> A: We have cited and discussed these works in Section 2 as follows:
>
> Wide local minima was considered an important regularization in CL to enforce the similarity of important parameters learned from past tasks [1]. Sharpness-aware seeking for loss landscape flat minima starts to gain more attention in CL, especially SAM based zeroth order sharpness is well discussed. An investigation [5] proves SAM can help with addressing forgetting in CL, and [4] proposed a combined SAM for few shot CL.
>
> SAM is also used for boosting the performance of specific methods like DFGP [2] and FS-DGPM [9] designed for GPM. SAM-CL [3] series with loss term gradient alignment for memory-based CL. These efforts kicked off the study of flat minima in CL, however, zeroth-order sharpness may not be enough for flatter optimal [62]. Thus, flatness with a global optima and universal CL framework is further studied.
>
> As for the connection and difference of C-Flat with various existing SAM methods, PLEASE SEE the beginning of Section 3 Method, where we have a very detailed theoretical explanation and analysis.
>
> >Weakness 3: Comparison with related work [1,2,3].
>
> A: The discussion is as follows,
>
> i) Note that C-Flat is an optimizer-level approach, thus, the performance gains on different CL-friendly optimizers should be compared rather than different CL methods. We have already done this in the initial manuscript, please see subsection 4.5/Table 2/Figure 5a/5b. And we further compared C-Flat with more sharpness-aware method, please see Table R4 and our response to Question 4 of Reviewer p3Vh.
>
> ii) Note that the flatness function used in DFGP is rooted in FS-GPM. We have demonstrated in Table 2 that C-Flat outperforms this function of FS-GPM.
>
> Therefore, the comparison with some related work you mentioned is out of the scope of this work. However, we still performed evaluations on CPR even if it was not a baseline against which we should compare. C-Flat still presents tangible gains, the results as below,
>
> | Method | Accuracy ↑ | Forgetting ↓ |
> |:--:|:--:|:--:|
> | Rwalk† |57.84| 9.37 |
> | w/ CPR† | 63.66 | 7.69  |
> | w/ C-Flat | **64.73** (+1.07) | **4.79** (-2.9) |
>
> >Weakness 4: Concern on computational cost.
>
> Actually, C-Flat excels in seeking well-generalized flat minima, which is demonstrated by the outstanding convergence speed (proved this in Figure 5a, and also see comments of Reviewer 475M). Second, C-Flat does not have to perform in every iteration, this means that repeated propagation can be significantly less. This suggests that C-Flat requires only a few epochs or iterations to converge to global optimum, like 50% of iterations or 60% of epochs (best result).
>
> Thus, in practice, we do not need to maintain the same settings as the baseline, which significantly enhances the practicality. Also, note that C-Flat is even faster than SGD in some cases while maintaining high performance, this is surely what the CL community expects. Thus, benefiting from the good optimization nature of C-Flat, this concern is trivial.
>
> Moreover, for easy use of C-Flat, we provided a tier guideline which categorizes C-Flat into L1 to L3 level. PLEASE SEE Weakness 3 of Reviewer 475M and our response to it.
>
> >Weakness 5: Slight performance gains in Table 2.
>
> A: i) Since the FS term in FS-GPM already visits the typical zeroth-order sharpness, at this point, C-Flat partly reconfigures FS-GPM via operate the curvature (Eq.3 and Eq.4). This implies that the gains of C-Flat are founded on the top of the typical sharpness optimization used. Such an experimental setup was intended to reaffirm the versatility of C-Flat. Hence, merely noting that the boost in Table 2 is slight, we do not think it is justified.
> ii) Since we set a fixed random seed during CL, the performance gain of C-Flat is fairly solid. This is rare enough for an optimizer-level effort. PLEASE SEE our response to Weakness 2 of Reviewer p3Vh and Table R1 about stability.
>
> >Question 1: About subtitle Gradient-based solution.
>
> A: Gradient-based solutions are a main group in CL, including shaping loss landscape, tempering the tug-of-war of gradient, and other learning dynamics [5, 20]. C-Flat seeks for stable CL by tuning the gradient to flat minima, which is quite different from most gradient-based CL methods. Therefore, we had this paragraph for comparison. We have added more context in line 80 to support the claim that gradient-based solutions focus on dynamic learning to overcome forgetting.
>
> >Question 2: Smooth loss curves after 80th epoch.
>
> A: The decay_steps was set to 80 for all methods, which indicates that the learning rate decay was performed after the 80th epoch. This caused a smooth loss curve.

---

> > ### Comment · Reviewer_PuVp · 2024-08-12
> >
> > Dear Authors,
> >
> > Thank you very much for your careful responses to each question, I read them in full.
> >
> > I also carefully read the comments of the other three reviewers and the author's responses. The other three reviewers' concerns focused on several other aspects, while my opinion mainly focused on "What are the significant contributions of this paper based on the existing flatness-aware CL-based methods?" In the response, the authors discuss differences and connections to existing methods and add comparisons to the CPR method. I hope the authors will add these latest discussions and results to the final version.
> >
> > In general, given that there have been multiple works on flatness-aware CL, the innovativeness of this paper will be slightly discounted from my perspective. However, given the theoretical analysis and efficiency optimizations provided by this paper, I am willing to change the score to a positive one.
> >
> >
> > Yours sincerely,
> >
> > Reviewer PuVp

---

> > > ### Author Response · Authors · 2024-08-12
> > >
> > > Thank you very much for your thorough review and thoughtful consideration of our work. We're grateful that you took the time to review the comments from the other reviewers as well. We will ensure that the latest discussions and results, including the comparisons to the CPR method, are incorporated into the version.
> > >
> > > Thank you for your valuable comments and for recognizing the efforts we have put into this work.

---

> ### Author Response · Authors · 2024-08-12
> **A Gentle Reminder of Feedbacks**
>
> Dear Reviewer PuVp,
>
> Thanks for your careful comments and your time for our work. We have revised our paper and added the discussion and experiments concerning
> + better clarification and tempered statements on contributions in the revised manuscript.
> + more thorough discussion and analysis with related work in the revised manuscript by citing the work you mentioned.
> + more thorough comparisons with the sharpness-aware optimization method in Table R4, with corresponding analysis in the revised manuscript.
> + the influence of computational efficiency, with corresponding analysis in the tier guideline section.
> + more thorough discussion about the performance gains in Table R1 and the Table from the response to Reviewer p3Vh.
>
> Currently, all of your concerns can be resolved in the revised version of the paper. We want to leave a gentle reminder that the discussion period is closing. We would appreciate your feedback to make sure that our responses and revisions have resolved your concerns, or whether there is a leftover concern that we can address to ensure a quality work.
>
>
> Yours sincerely,
>
> Authors of Paper 2181

---

### Official Review · Reviewer_p3Vh · 2024-07-12

**Soundness:** 3
**Presentation:** 3
**Contribution:** 3
**Rating:** 6
**Confidence:** 4

**Summary:**

The paper proposes a new algorithm agnostic optimisation method, which is tailored specifically for continual learning. This method takes advantage of zeroth order landscape sharpness-aware optimisation and proposes a new method, which improves upon SGD.

**Strengths:**

Originality: the method raises a question which is often overlooked, which is the role of stochastic optimisers in continual learning. While many of the existing methods focus on the continual learning algorithms, this submission considers orthogonal aspect, which is tailoring the optimisation process in accordance with the needs of continual learning.

Quality: the ablation studies show the parameter choice trade-off, time expenditure and the performance of the method; however the lack of confidence intervals does not allow to find out how significant this performance improvement is. The presentation and the derivation of the method look clear.

 Clarity: the paper is clear and well-structured in general, however, please refer to the weaknesses section for more detail on omissions.

**Weaknesses:**

Clarity: there has been a number of omissions in the paper which need to be fixed and which stay in a way of understanding of the paper. This includes, for example,

Line 38: "Another group of works seeks to preserve model generalization with **regulations** onto the training procedure itself" regularisation (instead of regulations?) of the training procedure?

Line 40: "are designed to encourage the training to efficiently extracting **feathers** for the current data space without forgetting" Features?

Line 188: 'Foster uses KL-based loss function to regularize the three combinations of old and new blocks for a stable performance on the previous data. It also introduces an effective redundant parameters and feature pruning strategy to maintain the single backbone model using knowledge distillation. DER follows the same framework, and introduces an auxiliary classifier and related loss item to encourage the model to learn diverse and discriminate features for novel concepts'
While I understand it is related to the methods described in Table 1, they are only discussed in the next section, it would be easier if the authors cited the FOSTER paper for clarity in  Section 3.1; furthermore, it is capitalised in the table but not in the description.

Quality: The proposed experimental analysis of the method does not come across as fully backing up the claims: from table 1, the accuracy improvement does not looks too big, especially given that there are no confidence intervals given which does not allow us to make the conclusions on whether the advantage is significant.

**Questions:**

1) the confidence intervals for the experimental  analysis is crucial for the understanding of the performance improvements.
2) The comparison is only done in relation to the standard SGD; would it make sense to compare such results with other optimisers such, e.g., Adam? Does this optimiser improve upon the SGD performance in the standard, offline learning scenarios?
3) The paper only evaluates the optimiser on class-incremental learning benchmarks. I wonder how does the method perform on domain-incremental settings using, for example, the protocol from van de Ven et al (2022)
4) Should the authors also compare the proposed method with other methods taking advantage of sharpness-aware optimisation?  If not, why?  The concern is that currently the model only compares with a weak baseline of standard SGD, while existing similar methods might provide better performance.

van de Ven, G.M., Tuytelaars, T. & Tolias, A.S. Three types of incremental learning. Nat Mach Intell 4, 1185–1197 (2022). https://doi.org/10.1038/s42256-022-00568-3

[21] Haowei He, Gao Huang, and Yang Yuan. Asymmetric valleys: Beyond sharp and flat local minima. NeurIPS, 32, 2019.

**Limitations:**

1) It would be important to hear more about the choice of parameters: Section 4.1 states that for all methods, the hyperparameters have been chosen to be the same. At the same time, it would be interesting to know what procedure did the authors use to choose these hyperparameters.

2) It might be also good to hear if the authors have any details on failure modes for the method.

---

> ### Author Rebuttal · Authors · 2024-08-06
>
> **NOTE**: Table R1-R4 and Figure R1 are attached to the one-page PDF.
>
> > Weakness 1: Fix the omissions.
>
> A: Thanks for your kind reminder. We have fixed the omissions and conducted a thorough proofreading for clarity.
>
> > Weakness 2: Significance analysis.
>
> A: i) Note that we set the fixed random seed (seed everything) to ensure that the training results are unique in all experiments during CL [58,59], which shows that our significance is well-supported.
>
> ii) Your nice suggestion inspires us that the class order (or task order) often significantly affects the performance of CL methods, a significance analysis of this is crucial. Hence, we set various seeds (1993, 1995, 1997) to split the dataset to further analyze the significance of C-Flat. In Table R1, we report the results, mean, std of running 3 times on different datasets. Table R1 concludes that the results of C-Flat remains significant across 3 datasets on all case. Finally, we will discuss significance in the revision.
>
> > Question 1: Compare with Adam.
>
> A: In Table R2, we further compare C-Flat to Adam. Table R2 shows that C-Flat improves the performance of Adam in most cases.
>
> Moreover, the reasons why we use SGD as a baseline in evaluation instead of Adam are as follows,
>
> + SGD is the mainstream used in the CL community[58,59]. In CL, SGD performs better than Adam in CNN based vision tasks with higher test accuracy. This can also be concluded from the result of SGD and Adam in Table 2. While Adam outperforms SGD in Transformer-based models, like the foundation model, language model.
>
> + Adam is better at saddle-point escaping but worse at flat minima selection than SGD[C2]. We analyze the mean escape time $\tau$ as follows,
>
> $\tau$ of SGD is $\log (\tau)=O\left(\frac{B \Delta L_{a b}}{\eta H_a}\right)$; $\tau$ of Adam is $\log (\tau)=O\left(\frac{\sqrt{B} \Delta L_{a b}}{\eta \sqrt{H_a}}\right)$
>
> $\tau$ of Adam exponentially depends on the square root of the eigenvalues of the Hessian at a minimum[C2], while $\tau$ of SGD exponentially depends on the eigenvalue of the Hessian at minima along the escape direction[C1].
>
> + The saddle-point escape behavior of Adam is approximately independent of saddle-point flatness, whereas SGD is linearly dependent[C2].
>
> Therefore, these observations and conclusions above motivated us to use SGD as a baseline.
>
> [C1] A diffusion theory for deep learning dynamics: Stochastic gradient descent exponentially favors flat minima. ICLR 2021.
>
> [C2] Adaptive Inertia: Disentangling the Effects of Adaptive Learning Rate and Momentum. ICML 2022.
>
> > Question 2: The performance in offline-learning.
>
> A: Adhere to the protocol [C3], we evaluate the effectiveness of C-Flat in offline learning scenes. As shown in Table R2, C-Flat still provides stable performance gains. This concludes that C-Flat still converges to flat minima even in offline learning cases, showing C-Flat features a good generalization.
>
> [C3] Supervised Contrastive Replay: Revisiting the Nearest Class Mean Classifier in Online Class-Incremental Continual Learning. CVPR 2021.
>
> > Question 3: The performance in domain-incremental settings.
>
> A: Indeed, it should be interesting to see the performance of C-Flat in domain-incremental tasks, and Reviewer 475M is wondering about that as well. We would be happy to conduct a further evaluation on C-Flat.
>
> So we tested the effectiveness of C-Flat according to the protocol [48]. Table R3 shows that C-Flat still provides consistent performance gains in domain-incremental scenarios. This means that C-Flat smoothly migrates to the global optimal of the joint space of the new and old tasks, thus mitigating catastrophic forgetting. This trend also occurs in task-incremental scenarios.
>
> Apart from the typical benchmark, inspired by your suggestion and Reviewer 475M, we further investigated how a larger domain gap affects C-Flat. In this case, the domain shifts much more drastically, thus posing a non-trivial challenge for C-Flat. We trained on ImageNet21K and then evaluated CL performance on IN-R/ObjNet. Please see the Table below, C-Flat surprisingly holds solid performance gains. Such an observation further enhances the technical contribution of C-Flat. You could also refer to Reviewer 475M's question and our response to it.
>
> | Method | IN-R B0_Inc20 | ObjNet B0_Inc10 |
> |:-:|:-:|:-:|
> | iCaRL | 72.13 | 48.06 |
> | w/ C-Flat  |  **72.92** (+0.79) | **49.59** (+1.53) |
> | MEMO | 70.96 | 56.22 |
> | w/ C-Flat | **71.69** (+0.73) | **56.50** (+0.28) |
>
> > Question 4: Compare C-Flat with other sharpness-aware optimization.
>
> A: Actually, sharpness-aware method is relatively new in CL, among the limited works, we took the more general approach FS-DGPM for comparison in Table 2.
> We compared the accuracy, convergence speed and result with different of parameters of C-Flat with other sharpness-aware method like SAM, GAM in Section 4.6 and 4.7 with Figure 5 and 6.
>
> To ensure the reliability, we further expanded the scope of the evaluation. In details, we compare C-Flat with more sharpness-aware methods, e.g, SAM, GSAM (newly added), and GAM. Then, the baseline methods used also increased from 1 of Figure 5 to 3, including Replay, WA, MEMO. Table R4 shows that C-Flat still achieves better accuracy.
>
> > Limitation 1: Choice of hyper-parameters.
>
> A: If not specified, the hyper-parameters for tested models are defaulted according to the open-source repo [58,59]. Each task are initialized with the same $\rho$ and $\eta$, which drops with iterations according to the scheduler described in section 4.7.
>
> > Limitation 2: Any details on failure modes for C-Flat.
>
> A: As our response to Q1, since the saddle-point escape behavior of Adam is approximate independent of saddle-point flatness, it cannot be as good as SGD at finding flat minima. In this sense, C-Flat is prone to fail on the Adam series, the results in Table R2 reconfirm this. As shown in Table R2, Adam showed a slight drop in some cases. We will discuss these in the revision.

---

> > ### Comment · Reviewer_p3Vh · 2024-08-09
> >
> > "A: i) Note that we set the fixed random seed (seed everything) to ensure that the training results are unique in all experiments during CL [58,59], which shows that our significance is well-supported."
> >
> > Not sure I can agree with it: the random seed might be the same but it won't address the concern that given the uncertainty of the random seed choice there may be a variation in the outcome. The same random seed might be somehow advantageous for one algorithm while being disadvantageous for the other. Choosing the same random seed does not replace the need in confidence intervals. Or am I missing anything?
> >
> > Checking the rebuttal, including to the other reviewers, in the meantime.

---

> ### Author Response · Authors · 2024-08-11
>
> We sincerely apologize for any misunderstanding regarding the significance you mentioned, and we appreciate your feedback on this matter. In this response, we have re-evaluated the uncertainty of our method using different random seeds. The updated results, including the mean and standard deviation over three runs across various benchmarks, are presented in the table below. As shown, our method continues to demonstrate significant and stable performance.
>
> Thank you for your patience and for bringing this to our attention, which is greatly appreciated.
>
> | Method      | CIFAR-100      | Tiny-ImageNet   |
> |:------------:|:--------------:|:--------------:|
> | Replay      | 58.58 (±0.39)   | 43.25 (±0.29)    |
> | w/ C-Flat   | 59.67 (±0.41)   | 45.22 (±0.26)    |
> | WA          | 66.52 (±0.17)   | 55.90 (±0.29)    |
> | w/ C-Flat   | 67.62 (±0.13)   | 56.43 (±0.29)    |
> | MEMO        | 69.72 (±0.30)   | 58.12 (±0.06)    |
> | w/ C-Flat   | 70.02 (±0.09)   | 58.57 (±0.33)    |
>
> We also summarized the detailed results for each seed in the below table,
>
> + The results with each seed on CIFAR-100 are as follows,
>
> | Method      | 1993  | 1995  | 1997  |
> |:-------------:|:------------:|:------------:|------------:|
> | Replay      | 58.87      | 58.83      | 58.03      |
> | w/ C-Flat   | **59.42**      | **60.25**      | **59.35**      |
> | WA          | 66.76      | 66.35      | 66.45      |
> | w/ C-Flat   | **67.79**      | **67.47**      | **67.60**      |
> | MEMO        | 69.82      | 69.31      | 70.03      |
> | w/ C-Flat   | **69.94**      | **69.98**      | **70.15**      |
>
> + The results with each seed on Tiny-ImageNet are as follows,
>
> | Method      | 1993 | 1995 | 1997 |
> |:-------------:|:---------------:|:---------------:|:---------------:|
> | Replay      | 43.31         | 42.87         | 43.56         |
> | w/ C-Flat   | **44.95**         | **45.58**         | **45.13**         |
> | WA          | 55.69         | 55.70         | 56.30         |
> | w/ C-Flat   | **56.06**         | **56.46**         | **56.78**         |
> | MEMO        | 58.15         | 58.04         | 58.17         |
> | w/ C-Flat   | **58.97**         | **58.16**         | **58.57**         |

---

> > ### Comment · Reviewer_p3Vh · 2024-08-11
> >
> > I've carefully checked the comments and the discussion with all the reviewers, and I think the authors addressed the concerns in the rebuttal. The confidence intervals show that the method improves upon the baseline in most of the scenarios. I also see that the authors have addressed the concerns about computational efficiency from the other reviewers. Therefore, I changed the score to recommend acceptance. I also think it would be expected that the code is released if the paper is accepted.
> >
> > However, one extra thing: I noticed that some of the confidence intervals are overlapping between the proposed method and the baseline, in this case I would not expect it to be highlighted in bold.

---

> > > ### Author Response · Authors · 2024-08-11
> > >
> > > Thank you for recognizing our work and response. We have corrected the bolded text for clarity and commit to releasing the code to the CL community.
> > >
> > > We appreciate your time and effort in reviewing our work and would like to express our respect for your responsible review.

---

### Author Rebuttal · Authors · 2024-08-07

We thank the reviewers for their valuable comments and appreciation of our strengths, e.g.,

+ well-presented and easy to follow (**Reviewer p3Vh/PuVp/Tb4q/475M**);
+ originality and nice novelty (**Reviewer p3Vh/Tb4q/475M**);
+ good generalizability for CL (**Reviewer p3Vh/ PuVp/ Tb4q/475M**).
+ well-supported theoretical and grounded work (**Reviewer Tb4q/475M**).

All suggestions are seriously considered in our rebuttal and we have carefully revised the manuscript to address the concerns.

**NOTE**: Table R1-R4 and Figure R1 are attached to the one-page PDF.

---

### Decision · Program_Chairs · 2024-09-25

**Decision:**

Accept (poster)

**Comment:**

This paper proposes C-Flat, a novel optimization method for continual learning (CL) that aims to find flatter minima in the loss landscape to mitigate catastrophic forgetting. The method is presented as a plug-and-play solution that can be integrated into various existing CL approaches. The authors also demonstrate that their method results in a performance boost in a variety of cases.

The review process highlighted a number of weaknesses such as not using multiple seeds for calculating the result significance but these were mostly addressed in the rebuttal resulting in a paper that the AC is happy to recommend Accept.